# PREDICTED VARIABLES IN PROGRAMMING

## ABSTRACT

We present *Predicted Variables* (PVars), an approach to making machine learning (ML) a first class citizen in programming languages. There is a growing divide in approaches to building systems: using human experts (e.g. programming) on the one hand, and using behavior learned from data (e.g. ML) on the other hand. PVars aim to make using ML in programming easier by hybridizing the two. We leverage the existing concept of variables and create a new type, a *predicted* variable. PVars are akin to native variables with one important distinction: PVars determine their value using ML when evaluated. We describe PVars and their interface, how they can be used in programming, and demonstrate the feasibility of our approach on three algorithmic problems: binary search, QuickSort, and caches. We show experimentally that PVars are able to improve over the commonly used heuristics and lead to a better performance than the original algorithms. As opposed to previous work applying ML to algorithmic problems, PVars have the advantage that they can be used within the existing frameworks and do not require the existing domain knowledge to be replaced. PVars allow for a seamless integration of ML into existing systems and algorithms. Our PVars implementation currently relies on standard Reinforcement Learning (RL) methods. To learn faster, PVars use the heuristic function, which they are replacing, as an *initial function*. We show that PVars quickly pick up the behavior of the initial function and then improve performance beyond that without ever performing substantially worse – allowing for a safe deployment in critical applications.

## 1 INTRODUCTION

Machine Learning (ML) has had many successes in the past decade in terms of techniques and systems as well as in the number of areas in which it is successfully applied. However, using ML has some cost that comes from the additional complexity added to software systems (Sculley et al., 2014). There is a fundamental impedance mismatch between the approaches to system building. Software systems have evolved from the idea that experts have full control over the behavior of the system and specify the exact steps to be followed. ML on the other hand has evolved from learning behavior by observing data. It allows for learning more complex but implicit programs leading to a loss of control for programmers since the behavior is now controlled by data. We believe it is very difficult to move from one to another of these approaches but that a hybrid between them needs to exist which allows to leverage both the developer's domain-specific knowledge and the adaptability of ML systems.

We present *Predicted Variables* (PVars) as an approach to hybridize ML with programming. We leverage the existing concept of a variable which is universal across all programming modalities and add a new type, a *predicted* variable. PVars are akin to native variables with one important distinction: a PVar determines its value using ML when evaluated. A developer will be able to use a PVar just like any other variable, combine it with heuristics, domain specific knowledge, problem constraints, etc. in ways that are fully under the programmer's control. This represents an *inversion of control* compared to how ML systems are usually built. PVars allow to integrate ML tightly into algorithms whereas traditional ML systems are build around the model.

PVars aim to make using ML in software development easier by avoiding the overhead of going through the traditional steps of building an ML system: (1) collecting and preparing training data, (2) defining a training loss, (3) training an initial model, (4) tweaking and optimizing the model, (5)

integrating the model into their system, and (6) continuously updating and improving the model to adjust for drift in the distribution of the data processed.

We show how these properties of PVars allow for applying ML in domains that have traditionally not been using ML. We demonstrate that ML can help improve the performance of "classical" algorithms that typically rely on a heuristic. The concrete implementation of PVars in this paper is based on standard deep reinforcement learning (RL). We emphasize that this is just one possible implementation. Other types of machine learning are in scope for PVars: supervised learning can be used when ground truth is available, or active learning is applicable when humans are in the loop to provide feedback.

While in this paper we show PVars in the context of the Python programming language and use concepts from object oriented programming, everything described here applies directly to functional or procedural programming languages as well. We describe the framework around the central concept of a *predicted variable* but depending on the language the notion of a *predicted function* can be used interchangeably.

We also introduce the notion of an *initial function* which can be the current heuristic that a PVar is replacing. It allows the PVar to minimize regret in critical applications and allow for safe deployment. This is a key strengths of our hybrid approach: it allows for better solutions while also providing better guarantees to the programmer.

We demonstrate the feasibility of our approach on three algorithmic problems: binary search, Quick-Sort, and caches where we replace and enrich commonly used heuristics. We show improvements over common heuristics by injecting a predicted variable into an existing program, leaving much of the algorithm (including the domain-specific knowledge) untouched. We consider these problems the first applications of our newly defined interface and see the main contribution of this paper in the general applicability of the framework. The problem selection in this paper was driven by the desire for a self-contained setup and ease of reproducibility. PVars are applicable to more general problems across a large variety of domains from system optimization to user modelling.

In our experiments we do not focus on the actual run time but rather on the effectiveness of the ML models. While for the algorithmic examples in this paper, in a practical scenario speed is the key metric, we see PVars as a more general interface that can be applied across a more diverse set of problems including user modelling, predicting user preference, or content recommendations. In many applications, speed is not a meaningful metric. Further, we believe that advances in specialized hardware will enable running machine learning models at insignificant cost (Kraska et al., 2018).

Our main contributions are:

- we introduce the PVar API to smoothly integrate ML into software development;
- we show how standard RL methods can be leveraged through the PVars interface;
- we propose an approach to learn using the initial function, leveraging off-policy learning;
- we demonstrate the feasibility of our approach on 3 standard algorithmic problems.

The remainder of this paper is structured as follows: We describe how PVars can be used in software development in sec. 2 and how we make use of the heuristics that we are replacing to guide the training and avoid unstable behavior in sec. 3. Sec. 4 describes our implementation and the application of PVars to three algorithmic applications. We also describe the experiments that we performed to demonstrate that PVars are an intuitive approach to bring ML closer to software development and are applicable to different problems. We describe related work in sec. 5.

## 2 SOFTWARE DEVELOPMENT WITH PVARS

A PVar has a simple API that allows the developer to provide enough information about its context, predict its value, and provide feedback about the quality of its predictions. PVars invert the control compared to common ML approaches that are model centric. Here, the developer has full control over how data and feedback are provided to the model, how inference is called, and how its results are used.

To create a PVar, the developer chooses its output type (float, int, category, ...), shape, and range; defines which data the PVar is able to observe (type, shape, range); and optionally provides an initial

function. In the following example we instantiate a scalar float PVar taking on values between $0$ and $1$, which can observe three scalar floats (each in the range between $0$ and $10$), and which uses a simple initial function:

```
pvar = PVar(
  output_def=(float,shape=[1],range=[0,1]),
  observation_defs={'low':(float,[1],[0,10]), 'high':(float,[1],[0,10]),
                    'target':(float,[1],[0,10])},
  initial_function=lambda observations: 0.5)
```

The PVar can then be used like a normal variable. It determines its value at read time by using inference in the underlying ML model, e.g.

```
value = pvar.Predict()
```

Specifically, developers should be able to use a PVar instead of a heuristic or an arbitrarily chosen constant. PVars can also take the form of a stochastic variable, shielding the developer from the underlying complexity of inference, sampling, and explore/exploit strategies.

The PVar determines its value on the basis of observations about the context that the developer passes in:

```
pvar.Observe('low', 0.12)
pvar.Observe({'high': 0.56, 'target': 0.43})
```

A developer might provide additional side-information into the PVar that an engineered heuristic would not be using but which a powerful model is able to use in order to improve performance.

The developer provides feedback about the quality of previous predictions once it becomes available:

```
pvar.Feedback(reward=10)
```

In this example we provide numerical feedback. Following common RL practice a PVar aims to maximize the sum of reward values received over time (possibly discounted). In other setups, we might become aware of the correct value in hindsight and provide the "ground truth" answer as feedback, turning the learning task into a supervised learning problem. Some problems might have multiple metrics to optimize for (run time, memory, network bandwidth) and the developer might want to give feedback for each dimension.

This API allows for integrating PVars easily and transparently into existing applications with little overhead. See listing 1 for how to use the PVar created above in binary search. In addition to the API calls described above, model hyperparameters can be specified through additional configuration, which can be tuned independently. The definition of the PVar only determines its interface (i.e. the types and shapes of inputs and outputs).

## 3  INITIAL FUNCTIONS IN PREDICTED VARIABLES

We allow for the developer to pass an initial function to the PVar. We anticipate that in many cases the initial function will be the heuristic that the PVar is replacing. Ideally it is a reasonable guess at what values would be good for the PVar to return. The PVar will use this initial function to avoid very bad performance in the initial predictions and observe the behavior of the initial function to guide its own learning process, similar to imitation learning (Hussein et al., 2017). The existence of the initial function should strictly improve the performance of a PVar. In the worst case, the PVar could choose to ignore it completely, but ideally it will allow the PVar to explore solutions which are not easily reachable from a random starting point. Further, the initial function plays the role of a heuristic policy which explores the state and action space generating initial trajectories which are then used for learning. Even though such exploration is biased, off-policy RL can train on this data. In contrast to imitation learning where an agent tries to become as good as the expert, we explicitly aim to outperform the initial function as quickly as possible, similar to Schmitt et al. (2018).

For a PVar to make use of the initial heuristic, and to balance between learning a good policy and the safety of the initial function, it relies on a *policy selection strategy*. This strategy switches between exploiting the learned policy, exploring alternative values, and using the initial function. It can be applied at the action or episode level depending on the requirements. Finally, the initial function provides a safety net: in case the learned policy starts to misbehave, the PVar can always fallback to the initial function with little cost.

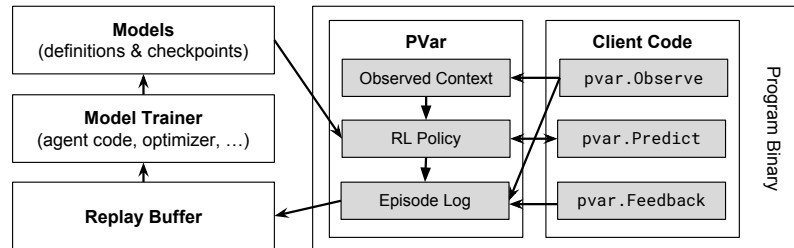

**Figure 1:** An overview of the architecture for our experiments how client code communicates with a PVar and how the model for the PVar is trained and updated.

## 4 APPLICATIONS OF PREDICTED VARIABLES IN ALGORITHMS

In the following we describe how PVars can be used in three different algorithmic problems and how a developer can leverage the power of machine learning easily with just a few lines of code. We show experimentally how using PVars helps improving the algorithm performance. The interface described above naturally translates into an RL setting: the inputs to **Observe** calls are combined into the state, the output of the **Predict** call is the action, and **Feedback** is the reward.

To evaluate the impact of PVars we measure **cumulative regret** over training episodes. Regret measures how much worse (or better when it is negative) a method performs compared to another method. Cumulative regret captures whether a method is better than another method over all previous decisions. For practical use cases we are interested in two properties: (1) Regret should never be very high to guarantee acceptable performance of the PVar under all circumstances. (2) Cumulative regret should become permanently negative as early as possible. This corresponds to the desire to have better performance than the baseline model as soon as possible.

Unlike the usual setting which distinguishes a training and evaluation mode, we perform evaluation from the point of view of the developer without this distinction. The developer just plugs in the PVar and starts running the program as usual. Due to the online learning setup in which PVars are operating, overfitting does not pose a concern (Dekel & Singer, 2005). The (cumulative) regret numbers thus do contain potential performance regressions due to exploration noise. This effect could be mitigated by performing only a fraction of the runs with exploration.

For our feasibility study we do not account for the computational costs of inference in the model. PVars would be applicable to a wide variety of problems even if these costs were high, particularly for problems relying on expensive approximation heuristics or working with inherently slow hardware, such as filesystems.

Our implementation currently is a small library exposing the PVar interface to client applications (fig. 1). A PVar assembles observations, actions, and feedback into episode logs that are passed to a replay buffer. The models are trained asynchronously. When a new checkpoint becomes available the PVar loads it for use in consecutive steps.

### 4.1 EXPERIMENT SETUP

To enable PVars we leverage recent progress in RL for modelling and training. It allows to apply PVars to the most general use cases. While we are only looking at RL methods here, PVars can be used with other learning methods embedded such as supervised learning or multi-armed bandit methods. We are building our models on DDQN (Hasselt et al., 2016) for categorical outputs and on TD3 (Fujimoto et al., 2018) for continuous outputs. DDQN is a de facto standard in RL since its success in AlphaGo (Silver et al., 2016). TD3 is a recent modification to DDPG (Lillicrap et al., 2015) using a second critic network to avoid overestimating the expected reward. We summarize the hyperparameters used in our experiments in the appendix (table 5). While these hyperparameters are now new parameters that the developer can tweak, we hypothesize that on the one hand, tuning hyperparameters is often simpler than manually defining new problem-specific heuristics and on the other hand that improvements on automatic model tuning from the general machine learning community will be easily applicable here too.

Our policy selection strategy starts by only evaluating the initial function and then gradually starts to increase the use of the learned policy. It keeps track of the received rewards of these policies adjusts

**Listing 1:** Standard binary search (left) and a simple way to use a PVar in binary search (right).

```
def bsearch(x, a, l=0, r=len(a)-1):       def bsearch(x, a, l=0, r=len(a)-1):
  if l > r: return None                     if l > r: return None
                                            pvar.Observe({'target':x,
                                              'low':a[l], 'high':a[r]})
  q = 0.5                                   q = pvar.Predict()
  m = int(q*l + (1-q)*r)                    m = int(q*l + (1-q)*r)
  if a[m] == x:                             if a[m] == x:
    return m                                  return m
                                            pvar.Feedback(-1)
  if a[m] < x:                              if a[m] < x:
    return bsearch(x, a, m+1, r)              return bsearch(x, a, m+1, r)
  return bsearch(x, a, l, m-1)             return bsearch(x, a, l, m-1)
```

the use of the learned policy depending on its performance. We show the usage rate of the initial function when we use it (fig. 2, bottom) demonstrating the effectiveness of this strategy.

Similar to many works that build on RL technology we are faced with the reproducibility issues described by Henderson et al. (2018). Among multiple runs of any experiment, only some runs exhibit the desired behavior, which we report. However, in the "failing" runs we observe baseline performance because the initial function acts as a safety net. Thus, our experiments show that we can outperfom the baseline heuristics without a high risk to fail badly. We do not claim to have a solution to these reproducibility issues but any solution developed by the community will be applicable here. To quantify the reproducibility of our results for the different problems, we provide the performance of the learned policies in the appendix when re-running the same experiments multiple times.

## 4.2 BINARY SEARCH

Binary search (Williams, 1976) is a standard algorithm for finding the location $l_x$ of a target value $x$ in a sorted array $A = \{a_0, a_1, \ldots, a_{N-1}\}$ of size $N$. Binary search has a worst case runtime complexity of $\lceil \log_2(N) \rceil$ steps when no further knowledge about the distribution of data is available. Knowing more about the distribution of the data can help to reduce expected runtime. For example, if the array values follow a uniform distribution, the location of $x$ can be approximated using linear interpolation $l_x \approx (N-1)(x-a_0)/(a_{N-1}-a_0)$. We show how PVars can be used to speed up binary search by learning to estimate the position $l_x$ for a more general case.

The **simplest way** of using a PVar is to directly estimate the location $l_x$ and incentivize the search to do so in as few steps as possible by penalizing each step by the same negative reward (listing 1). At each step, the PVar observes the values $a_L$, $a_R$ at both ends of the search interval and the target $x$. The PVar output $q$ is used as the relative position of the next read index $m$, such that $m = qL + (1-q)R$.

In order to give a **stronger learning signal** to the model, the developer can incorporate problem-specific knowledge into the reward function or into how the PVar is used. One way to **shape the reward** is to account for problem reduction. For binary search, reducing the size of the remaining search space will speed up the search proportionally and should be rewarded accordingly. By replacing the step-counting reward in listing 1 (line 9) with the search range reduction $(R_t - L_t)/(R_{t+1} - L_{t+1})$, we directly reward reducing the size of the search space. By shaping the reward like this, we are able to attribute the feedback signal to the current prediction and to reduce the problem from RL to contextual bandit (which we implement by using a discount factor of 0).

Alternatively we can **change the way the prediction is used** to cast the problem in a way that the PVar learns faster and is unable to predict very bad values. For many algorithms (including binary search) it is possible to predict a combination of (or choice among) several existing heuristics rather than predicting the value directly. We use two heuristics: (a) vanilla binary search which splits the search range $\{a_L, \ldots, a_R\}$ into two equally large parts using the split location $l^v = (L + R)/2$, and (b) interpolation search which interpolates the split location as $l^i = ((a_R - v)L + (v - a_L)R)/(a_R - a_L)$. We then use the value $q$ of the PVar to mix between these heuristics to get the predicted split position $l^q = ql^v + (1-q)l^i$. Since in practice both of these heuristics work well on many distributions, any point in between will also work well. This reduces the risk for the PVar to pick a value that is really bad which in turn helps learning. A disadvantage is that it's impossible to find the optimal strategy with values outside of the interval between $l^v$ and $l^i$.

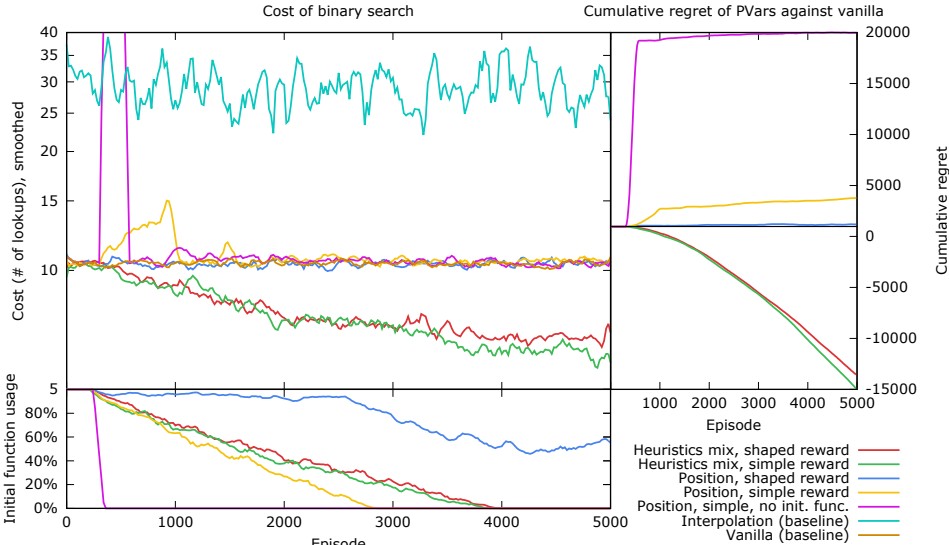

**Figure 2:** The cost of different variants of binary search (top left), cumulative regret compared to vanilla binary search (right), and initial function usage (bottom).

**Listing 2:** A QuickSort implementation that uses a PVar to choose the number of samples to compute the next pivot. As feedback, we use the cost of the step compared to the optimal partitioning.

```
def qsort(a, l=0, r=len(a)):          def pivot(a, l, r):
  if r <= l+1:                          pvar.Observe({'left':l, 'right':r})
    return                              q = min(1+2*pvar.Predict(), r-l)
  m = pivot(a, l, r)                    v = median(sample(a[l:r], q))
  qsort(a, l, m-1)                      m = partition(a, l, r, v)
  qsort(a, m+1, r)                      c = cost_of_median_and_partition()
                                        d = delta_cost(c, r-l, m-l, r-m)
def delta_cost(c_pivot, n, a, b):       pvar.Feedback(1/d)
  # See eq. 1                           return m
```

To **evaluate our approaches** we are using a test environment where in each episode, we sample an array of 5000 elements from a randomly chosen distribution (uniform, triangular, normal, pareto, power, gamma and chisquare), sort it, scale to $[-10^4, 10^4]$ and search for a random element.

Figure 2 shows the results for the different variants of binary search using a PVar and compares them to the vanilla binary search baseline. The results show that the simplest case (pink line) where we directly predict the relative position with the simple reward and without using an initial function performs poorly initially but then becomes nearly as good as the baseline (cumulative regret becomes nearly constant after an initial bad period). The next case (yellow line) has an identical setup but we are using the initial function and we see that the initial regret is substantially smaller. By using the shaped reward (blue line), the PVar is able to learn the behavior of the baseline quickly. Both approaches that are mixing the heuristics (green and red lines) significantly outperform the baselines.

In the appendix (table 1) we give details about when each of the different variants of using a PVar in binary search reaches break-even.

## 4.3 QUICKSORT

QuickSort (Hoare, 1962) sorts an array in-place by partitioning it into two sets (smaller/larger than the pivot) recursively until the array is fully sorted. QuickSort is one of the most commonly used sorting algorithms where many heuristics have been proposed to choose the pivot element. While the average time complexity of QuickSort is $\theta(N \log(N))$, a worst case time complexity of $O(N^2)$ can happen when the pivot elements are badly chosen. The optimal choice for a pivot is the median of the range, which splits it into two parts of equal size.

To improve QuickSort using a PVar we aim at tuning the pivot selection heuristic. To allow for sorting arbitrary types, we decided to use the PVar to determine the number of elements that are sampled from the array to be sorted and then pick the median from these samples as the pivot (listing 2). As

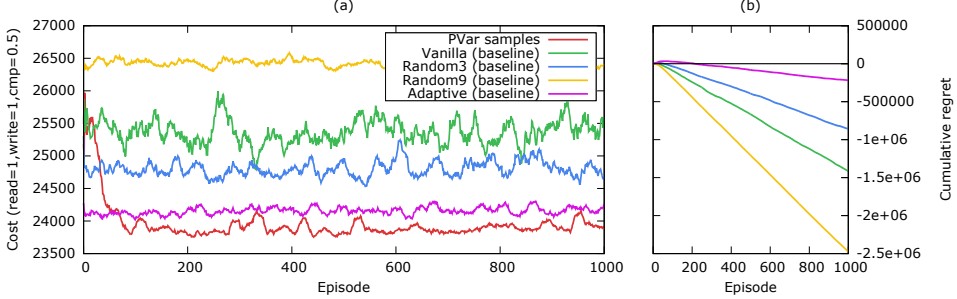

**Figure 3:** Results from using a PVar for selecting the number of pivots in QuickSort. (a) shows the overall cost for the different baseline methods and for the variant with a PVar over training episodes. (b) shows the cumulative regret of the PVar method compared to each of the baselines over training episodes.

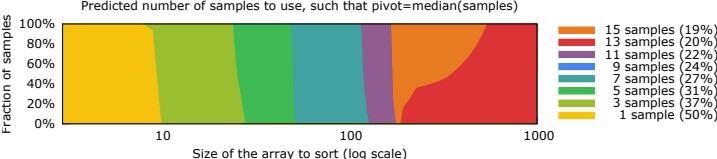

**Figure 4:** Fraction of pivots chosen by the PVar in QuickSort after 5000 episodes. The expected approximation error of the median is given in the legend, next to the number of samples.

**feedback signal** for a recursion step we use an estimate of its impact on the computational cost $\Delta c$.

$$\Delta c = \frac{c_{\text{pivot}} + \Delta c_{\text{recursive}}}{c_{\text{expected}}} = \frac{c_{\text{pivot}} + (a \log a + b \log b - 2 \frac{n}{2} \log \frac{n}{2})}{n \log n}, \quad (1)$$

where $n$ is the size of the array, $a$ and $b$ are the sizes of the partitions with $n = a + b$ and $c_{\text{pivot}} = c_{\text{median}} + c_{\text{partition}}$ is the cost to compute the median of the samples and to partition the array. $\Delta c_{\text{recursive}}$ takes into account how close the current partition is to the ideal case (median). The cost is a weighted sum of number of reads, writes, and comparisons. Similar to the shaped reward in binary search, this reward allows us to reduce the RL problem to a contextual bandit problem and we use a discount of 0.

For **evaluation** we are using a test environment where we sort randomly shuffled arrays. Results of the experiments are presented in fig. 3. It shows that the learned method outperforms all baseline heuristics within less than 100 episodes. 'Vanilla' corresponds to a standard QuickSort implementation that picks one pivot at random in each step. 'Random3' and 'Random9' sample 3 and 9 random elements respectively and use the median of these as pivots. 'Adaptive' uses the median of $\max(1, \lfloor \log_2(n) - 1 \rfloor)$ randomly sampled elements as pivot when partitioning a range of size $n$. It uses more samples at for larger arrays, leading to a better approximation of the median, and thus to faster problem size reduction.

Fig. 4 shows that the **PVar learns a non-trivial policy**. The PVar learns to select more samples at larger array sizes which is similar to the behavior that we hand-coded in the adaptive baseline but in this case no manual heuristic engineering was necessary and a better policy was learned. Also, note that a PVar-based method is able to adapt to changing environments which is not the case for engineered heuristics. One surprising result is that the PVar prefers 13 over 15 samples at large array sizes. We hypothesize this happens because relatively few examples of large arrays are seen during training (one per episode, while arrays of smaller sizes are seen multiple times per episode).

## 4.4 CACHES

Caches are a commonly used component to speed up computing systems. They use a *cache replacement policy* (CRP) to determine which element to evict when the cache is full and a new element needs to be stored. Probably the most popular CRP is the *least recently used* (LRU) heuristic which evicts the element with the oldest access timestamp. A number of approaches have been proposed to improve cache performance using machine learning (see sec. 5). We propose two different approaches how PVars can be used in a CRP to improve cache performance.

**Discrete** (listing 3): A PVar directly predicts which element to evict or chooses not to evict at all (by predicting an invalid index). That is, the PVar learns to become a CRP itself. While this is the

**Listing 3:** Cache replacement policy directly predicting eviction decisions (*Discrete*).

```
keys = ...  # keys now in cache.

# Returns evicted key or None.
def miss(key):
  pvar.Feedback(-1) # Miss penalty.
  pvar.Observe('access', key)
  pvar.Observe('memory', keys)
  return evict(pvar.Predict())

def evict(i):
  if i >= len(keys): return None
  pvar.Feedback(-1) # Evict penalty.
  pvar.Observe('evict', keys[i])
  return keys[i]
def hit(key):
  pvar.Feedback(1) # Hit reward.
  pvar.Observe('access', key)
```

**Listing 4:** Cache replacement policy using a priority queue (*Continuous*).

```
q = min_priority_queue(capacity)
def priority(key):
  pvar.Observe(...)
  score = pvar.Predict()
  score *= capacity * scale
  return time() + score

def hit(key):
  pvar.Feedback(1) # Hit reward.
  q.update(key, priority(key))
def miss(key):
  pvar.Feedback(-1) # Miss penalty.
  return q.push(key, priority(key))
```

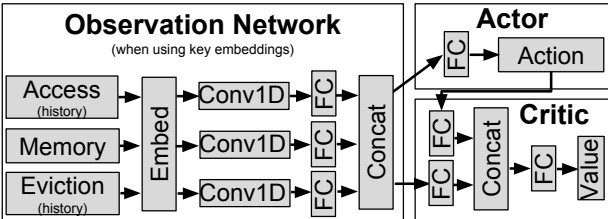

**Figure 5:** The architecture of the neural networks for TD3 with key embedding network.

simplest way to use a PVar, it makes it more difficult to learn a CRP better than LRU (in fact, even learning to be on par with LRU is non-trivial in this setting).

**Continuous** (listing 4): A PVar is used to enhance LRU by predicting an offset to the last access timestamp. Here, the PVar learns which items to keep in the cache longer and which items to evict sooner. In this case it becomes trivial to be as good as LRU by predicting a zero offset. The PVar value in $(-1, 1)$ is scaled to get a reasonable value range for the offsets. It is also possible to choose not to store the element by predicting a sufficiently negative score.

In both approaches the feedback given to the PVar is whether an item was found in the cache $(+1)$ or not $(-1)$. In the discrete approach we also give a reward of $-1$ if the eviction actually takes place.

In our implementation the observations are the history of accesses, memory contents, and evicted elements. The PVar can observe (1) keys as a categorical input or (2) features of the keys.

Observing **keys as categorical input** allows to avoid feature engineering and enables directly learning the properties of particular keys (e.g. which keys are accessed the most) but makes it difficult to deal with rare and unseen keys. To handle keys as input we train an embedding layer shared between the actor and critic networks (fig. 5).

As **features of the keys** we observe historical frequencies computed over a window of fixed size. This approach requires more effort from the developer to implement such features, but pays off with better performance and the fact that the model does not rely on particular key values.

We experiment with three combinations of these options: (1) discrete caches observing keys, (2) continuous caches observing keys, (3) continuous caches observing frequencies. For **evaluation** we use a cache with size 10 and integer keys from 1 to 100. We use two synthetic access patterns of length 1000, sampled i.i.d. from a power law distribution with $\alpha = 0.1$ and $\alpha = 0.5$. Fig. 6 shows results for the three variants of predicted caches, a standard LRU cache, and an oracle cache to give a theoretical, non-achievable, upper bound on the performance.

We look at the hit ratio without exploration to understand the potential performance of the model once learning has converged. However, cumulative regret is still reported under exploration noise.

Both implementations that work directly on key embeddings learn to behave similar to the LRU baseline without exploration (comparable hit ratio). However, the continuous variant pays a higher penalty for exploration (higher cumulative regret). Note that this means that the continuous variant

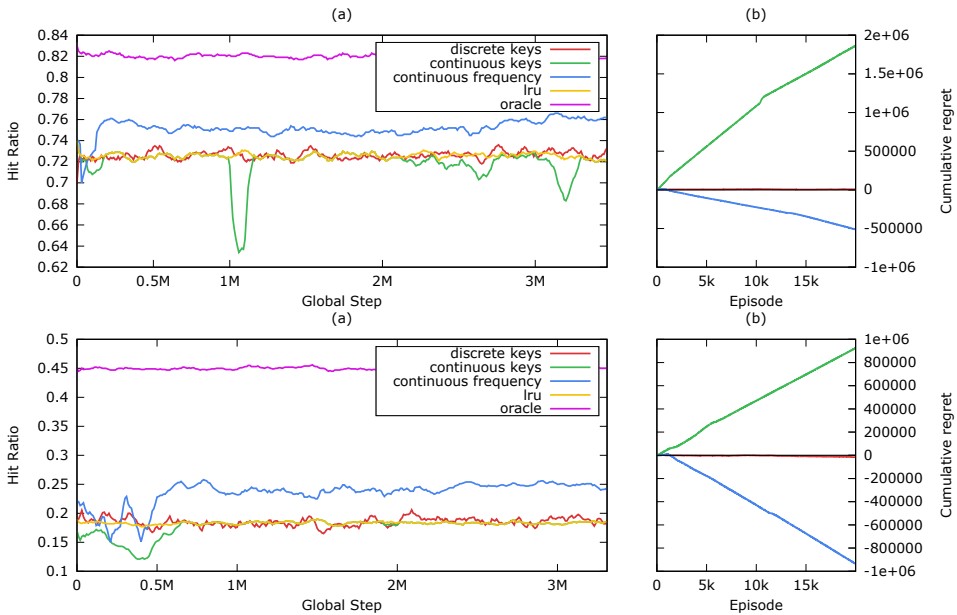

**Figure 6:** Cache performance for power law access patterns. Top: $\alpha = 0.1$, bottom: $\alpha = 0.5$. (a) Hit Ratio (w/o exploration) and (b) Cumulative Regret (with exploration)

learned to predict constant offsets (which is trivial), however the discrete implementation actually learned to become an LRU CRP which is non-trivial. The continuous implementation with frequencies quickly outperforms the LRU baseline, making the cost/benefit worthwhile long-term (negative cumulative regret after a few hundred episodes).

## 5 RELATED WORK

Similar to our proposed interface, probabilistic programming (Gordon et al., 2014) introduces interfaces which simplify the developer complexity when working with statistical models and conditioning variable values on run-time observations. In contrast to PVars, the introduced interfaces are specialized on working with distributions and graphical models. In the space of approximate computing, (Sampson et al., 2011) propose a programming interface for approximate computation. While similar in spirit, this work does not explicitly target machine learning models.

Similar in spirit to our approach is (Kraska et al., 2018) which proposes to incorporate neural models into database systems by replacing existing index structures with neural models that can be both faster and smaller. PVars in contrast aim not to replace existing data structures or algorithms but transparently integrate with standard algorithms and systems. PVars are general enough to be used to improve the heuristics in algorithms (as done here), to optimize database systems (similar to Kraska et al. (2018)), or to simply replace an arbitrarily chosen constant. Another approach that is similar to PVars is Spiral (Bychkovsky et al., 2018) but it is far more limited in scope than PVars in that it aims to predict boolean values only and relies on ground truth data for model building.

Similarly, a number of papers apply machine learning to algorithmic problems, e.g. Neural Turing Machines (Graves et al., 2014) aims to build a full neural model for program execution. Kaempfer & Wolf (2018); Kool et al. (2018); Bello et al. (2016) propose end-to-end ML approaches to combinatorial optimization problems. In contrast to PVars these approaches replace the existing methods with an ML-system. These are a good demonstration of the inversion of control mentioned above: using ML requires to give full control to the ML system.

There are a few approaches that are related to our use of the initial function, however most common problems where RL is applied do not have a good initial function. Generally related is the idea of imitation learning (Hussein et al., 2017) where the agent aims to replicate the behavior of an expert. Typically the amount of training data created by an expert is very limited. Based on imitation learning is the idea to use previously trained agents to kickstart the learning of a new model (Schmitt et al., 2018) where the authors concurrently use a teacher and a student model and encourage the

student model to learn from the teacher through an auxiliary loss that is decreased over time as the student becomes better.

In some applications it may be possible to obtain additional training data from experts from other sources, e.g. (Hester et al., 2018; Aytar et al., 2018) leverage YouTube videos of gameplay to increase training speed of their agents. These approaches work well in cases where it is possible to leverage external data sources.

Caches are an interesting application area where multiple teams have shown in the past that ML can improve cache performance (Zhong et al., 2018; Lykouris & Vassilvitskii, 2018; Hashemi et al., 2018; Narayanan et al., 2018; Gramacy et al., 2002). In contrast to our approach, all ML models work on caches specifically and build task-dependent models that do not generalize to other tasks.

Algorithm selection has been an approach to apply RL for improving sorting algorithms (Lagoudakis & Littman, 2000). Search algorithms have also been improved using genetic algorithms to tweak code optimization (Li et al., 2005).

# 6 CONCLUSION

We have introduced a new programming concept called a predicted variable (PVar) aiming to make it easier for developers to use machine learning from their existing code. Contrary to other approaches, PVars can easily be integrated and hand full control to the developer over how ML models are used and trained. PVars bridge the chasm between the traditional approaches of software systems building and machine learning modeling and thus allow for the developer to focus on refining their algorithm and metrics rather than working on building pipelines to incorporate machine learning. PVars achieve this by reusing the existing concept of variables in programming in a novel way where the value of the variable is determined using machine learning. PVar observes information about its context and receives feedback about the quality of predictions instead of being assigned a value directly.

We have studied the feasibility of PVars in three algorithmic problems. For each we show how easy PVars can be incorporated, how performance improves in comparison to not using a PVar at all. Specifically, through our experiments we highlight both advantages and disadvantages that reinforcement learning brings when used as a solution for a generic interface as PVars.

Note that we do *not* claim to have the best possible machine learning model for each of these problems but our contribution lies in building a framework that allows for using ML easily, spreading its use, and improving the performance in places where machine learning would not have been used otherwise. PVars are applicable to more general problems across a large variety of domains from system optimization to user modelling. Our current implementation of PVars is built on standard RL methods but other ML methods such as supervised learning are in scope as well if the problem is appropriate.

## FUTURE WORK

In this paper we barely scratch the surface of the new opportunities created with PVars. The current rate of progress in ML will enable better results and wider applicability of PVars to new applications. We hope that PVars will inspire the use of ML in places where it has not been considered before.

We plan to release the code to reproduce the results in this paper. Further, we hope to make PVars a standard feature in C++29, Python 4, and Java 12. ;)

## ACKNOWLEDGMENTS

The authors would like to thank George Baggott, Gabor Bartok, Jesse Berent, Andrew Bunner, Sergio Guadarrama, Efi Kokiopoulou, Anoop Korattikara , Eugene Kripichov, Ketan Mandke, Rif Sauros, Luciano Sbaiz, and Weikang Zhou for discussions and support.

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

## A BREAK-EVEN FOR BINARY SEARCH VARIANTS

Table 1 gives details about when each of the different variants of using a PVar in binary search reaches break-even. The numbers indicate how many episodes it takes for the cumulative regret to become permanently negative, which means that for any additional evaluations after that point the user has a net benefit from using a PVar compared to not using ML at all. The table shows that reward shaping and using the predictions smartly improve performance but it also shows that even simple methods are able to give improvements. Note, that no model outperforms interpolation search on a uniform distribution as it is the best approximation for this distribution.

**Table 1:** Training episodes required for the cumulative regret to become permanently negative (compared to all baselines) for all combinations of Prediction, Reward, and use of initial functions ("–": does not happen within 5000 episodes).

| Prediction | Position | | | | Heuristics Mix | | | |
|---|---|---|---|---|---|---|---|---|
| Reward | simple | | shaped | | simple | | shaped | |
| Initial function | no | yes | no | yes | no | yes | no | yes |
| Random | – | – | – | – | 1058 | – | 425 | **258** |
| Chisquare | – | – | 3231 | 4885 | 3937 | 285 | 409 | **240** |
| Gamma | – | – | 3218 | – | – | **248** | 594 | – |
| Normal | – | – | 3396 | – | 1048 | 283 | 403 | **252** |
| Pareto | – | – | 4255 | 4586 | – | 398 | 508 | **256** |
| Power | – | – | – | – | 1053 | – | 1234 | **234** |
| Triangular | – | – | – | – | **519** | 2618 | 666 | 2291 |
| Uniform | – | – | – | – | – | – | – | – |

## B REPRODUCIBILITY: GOALS AND METRICS

We do not claim to have solved reinforcement learning reproducibility and throughout our experiments we are facing the same issues as the larger community. The core aspect of the PVars framework is the ability to rely on the initial function or algorithmic formulation to limit the costs of a

poorly performing learned policy. We illustrate this by looking at metrics over 100 restarts of the different experiments and highlight that, while some experiments for some problems are more reproducible than others, we do not perform worse than the initial function provided by the developer.

The design construct specific to PVars and what distinguishes it from standard Reinforcement Learning is that it is applied in software control where often developers are able to provide safe initial functions or write the algorithm in a way that limits the cost of a poorly performing policy.

## B.1 BINARY SEARCH

To quantify the reproducibility, we ran the experiment from Sec. 4.2 120× and report the cumulative regret per episode (average number of extra steps per search episode) compared to vanilla binary search. On average, the cumulative regret is: -1.59 @5K (-2.20 @50K). The break-even point is reached in 85% of the cases, and within an average of 1965 episodes. The performance breakdown by percentile, and the number of steps at which the break-even point is reached are referenced in table 2.

**Table 2:** Binary Search reproducibility: average regret per episode (lower is better) and break-even point

| Percentile | 1 | 5 | 10 | 25 | 50 | 75 | 90 | 95 | 99 |
|---|---|---|---|---|---|---|---|---|---|
| Regret @5K episodes | -2.71 | -2.66 | -2.62 | -2.45 | -2.03 | -1.01 | 0.44 | 0.70 | 0.78 |
| Regret @50K episodes | -3.99 | -3.83 | -3.76 | -3.64 | -3.34 | -2.85 | 3.80 | 3.86 | 3.92 |
| Break-even point (episodes) | 127 | 201 | 271 | 417 | 758 | 2403 | $\infty$ | $\infty$ | $\infty$ |

## B.2 QUICKSORT

To quantify the reproducibility, we ran the experiment described in Sec. 4.3 115× and report the cumulative regret per episode (average number of extra operations, as read=write=1, compare=0.5 per sort) compared to vanilla QuickSort. On average, the cumulative regret per episode is -913 @1K (-1064 @10K) on a total operation cost of 25.1K per sort. The break-even point is reached in 94% of the cases, and in an average after 368 episodes. The performance breakdown by percentile, and the number of steps at which the break-even point is reached are referenced in table 3.

**Table 3:** QuickSort reproducibility: average regret per episode (lower is better) and break-even point

| Percentile | 1 | 5 | 10 | 25 | 50 | 75 | 90 | 95 | 99 |
|---|---|---|---|---|---|---|---|---|---|
| Regret @1K episodes | -1273 | -1248 | -1214 | -1146 | -1029 | -916 | -409 | 372 | 425 |
| Regret @10K episodes | -1356 | -1306 | -1267 | -1219 | -1146 | -1034 | -945 | -285 | 393 |
| Break-even point (episodes) | 0 | 0 | 0 | 37 | 93 | 141 | 307 | 7370 | $\infty$ |

## B.3 CACHES

In order to quantify the reproducibility of our experiments we ran 100 times the same experiment illustrated in sec. 4.4 and we report the performance of the learned cache policy using predicted variables when compared with LRU heuristic, by looking at the cumulative regret metric after 20000 episodes. We break down the cumulative regret by percentiles in table 4.

**Table 4:** Caches reproducibility: average regret per episode (lower is better) and break-even point

| Percentile | 1 | 5 | 10 | 25 | 50 | 75 | 90 | 95 | 99 |
|---|---|---|---|---|---|---|---|---|---|
| Regret @20K episodes | -8.25 | -5.88 | -3.49 | -0.00 | 0.00 | 0.02 | 0.34 | 0.84 | 2.17 |
| Break even point (episodes) | 32 | 157 | 472 | $\infty$ | $\infty$ | $\infty$ | $\infty$ | $\infty$ | $\infty$ |

When counting the number of runs for which there exists an episode where the cumulative regret is strictly negative until the end, we note that this happens for 26% of the runs. For 60% of the runs the cumulative regret does not become positive, meaning that using the learned cache policy is at least as good as using the LRU heuristic. This leaves us with 14% of the runs resulting in strictly worse performance than relying on the LRU heuristic.

## C  TD3, DDQN HYPERPARAMETERS

In table 5 we provide the hyperparameters used for different experiments in order to ease reproducibility of our work. Together with Sec. B this details our entire experimental results and setup.

**Table 5:** Parameters for the different experiments described below (FC=fully connected layer, LR=learning rate). See (Henderson et al., 2018) for details on these parameters.

| | Binary search | QuickSort | Caches (discrete) | Caches (continuous) |
|---|---|---|---|---|
| Learning algorithm | TD3 | DDQN | DDQN | TD3 |
| Actor network | $FC_{16} \to \tanh$ | – | – | $FC_{10} \to \tanh$ |
| Critic/value network | $FC_{16}$ | $(FC_{16}, ReLU)^2 \to FC$ | $(FC_{10}, ReLU)^2 \to FC$ | $FC_{10}$ |
| Key embedding size | – | – | 8 | |
| Discount | 0.8, 0 | 0 | 0.8 | |
| LR actor | $10^{-3}$ | – | – | $10^{-4}$ |
| Initial function decay | yes | no | | |
| Batch size | 256 | | 1024 | |
| Action noise $\sigma$ | 0.03 | – | – | 0.01 |
| Target noise $\sigma$ | 0.2 | – | – | 0.01 |
| Temperature | – | 0.1 | | – |
| Update ratio ($\tau$) | 0.05 | 0.001 | | |
| Common: Optimizer: Adam; LR critic: $10^{-4}$; Replay buffer: Uniform, FIFO, size 20000; Update period: 1. | | | | |

