# OpenReview forum: "Predicted Variables in Programming"
_ICLR.cc/2019/Conference_

### Official Review · AnonReviewer3 · 2018-10-30
**Interesting idea but replaces constants with other constants**

**Rating:** 7
**Confidence:** 3

**Review:**

The paper proposes to include within regular programs, learned parameters that are then tuned in an online manner whenever the program is invoked. Thus learning is continuous, integration with the ML backend seamless. The idea is very interesting however, it seems to me that while we can replace native variables with learned parameters, the hyperparameters involved in the learning become new native variables (e.g. the value of feedback). Perhaps with some effort we can replace the  hyperparameters with predicted variables too. Other concerns of mine stem from the programmer in me. I think of a program as something deterministic and predictable. With continuous, online, self-tuning, these properties are gone. How do the authors propose to assuage folks with my kind of mindset? Is debugging programs with predicted variables an issue? Consider a situation where the program showed some behavior with a certain setting of q which has since been tuned to another value and thus the same behavior doesn't show up. I find these to be very interesting questions but don't see much of a discussion in the current draft. Also, how does this work relate to probabilistic programming?

---

> ### Author Response · Authors · 2018-11-19
> **Addressed some of the questions through comments and updated submission.**
>
> We thank the reviewer for the comments and questions brought up related to our proposed interface.
>
> [hyperparameters become new variables]
> We agree that hyperparameters introduce an additional search space, but we consider that navigating through this space is sometimes simpler than in the space of building complex heuristic functions to improve a specific problem, which would be the equivalent of not being able to use machine learning at all through an interface such as PVars.
>
> [debugging programs with predicted variables]
> As with debugging any complex ML model, predicted variables will likely add additional challenges to debugging. However, because of their natural integration into the programming language, debugging the logic around the predicted variable should not be affected and inspecting the values coming from a predicted variable in a debugger will also be as simple as inspecting a regular predicted variable.
>
> [relation to probabilistic programming]
> Probabilistic programming is a line of work similar to ours but focused on a specific class of models. The interface introduced by the probabilistic programming line of work exposes directly methods required for operating with that class of models, e.g. graphical models, where as PVars leaves that as a solution detail.
> We added related work from the probabilistic programming literature to our paper.
>
> [interesting questions that aren't discussed much in the current draft]
> We have updated our draft to highlight our position related to some of your questions.
> We consider this work on predicted variables a first step into an interesting field of research and we hope to be able to address more of these questions in future work.

---

### Official Review · AnonReviewer1 · 2018-11-02
**Interesting proposal without clear contributions**

**Rating:** 5
**Confidence:** 3

**Review:**

This paper proposes the use of RL as a set of commands to be included as programming instructions in  common programming languages.  In this aspect, the authors propose to add simple instructions to employ the power of machine learning in general, and reinforcement learning in particular in common programming tasks.

In this aspect, the authors show with three different examples how the use of RL can speed up the performance of common tasks: binary search, sorting and caches.

The paper is easy to read and follow.

In my opinion, the main problem of the paper is that the contributions are not clear. The authors claim that the introduce a new hybrid approach of programming between common programming and ML, however, I do not see many differences between calling APIs and the current proposal. The paper seems to be a wrapper of API calls. Here, the authors should comment existing approaches based on ML and APIs.

The authors  introduce the examples to show the advantages of using predictive variables. Many of the advantages are based on increasing the performance of the algorithms using these predictive variables, however, the results do not include the computational costs of learning the models.

Therefore, in my opinion the paper should be more focused on detailing the commands of use of predictive variables and emphasising the advantages with respect to existing methods. Currently, the paper gives too relevance to the performance of the experiments, where the novel contributions are not there.

---

> ### Author Response · Authors · 2018-11-19
> **Clarified contributions in the paper**
>
> We thank the reviewer for their insightful comments and the very relevant question about clarifying our contributions. We have tried to clarify and itemized our contribution (see page 2).
>
> [no operational cost given]
> The main focus of the paper is not to improve specific algorithms but demonstrate that such improvement is possible easily, and illustrate the claim with simple/well-known algorithms examples.
>
> We did not provide an analysis of the computational overhead of our method because we see the three algorithmic problems as tasks to demonstrate that the interface that we provide is expressive and powerful enough to bring ML into normal software development. In many other applications, where predicted variables can be applied, speed is not a relevant metric, e.g user modelling , optimizing UI components, predicting user preference,  systems optimization, or content recommendations. We acknowledge that our current implementation is probably slower than the original variant - but as we describe above,  we don't consider actual runtime to be the relevant metric here.
> Further - we strongly believe that specialized hardware such as GPUs or TPUs are continuously improving the runtime of ML models which will eventually make our proposed implementation practical even for speed sensitive applications (compare also Kraska et al, 2017).
>
> ["commands of use"]
> We do agree with R2 that the main contribution of this paper is in the novel API that we propose. As we describe in the paper, the experiments are performed to demonstrate that such an API is actually feasible and to indicate how good the state of the art in machine learning supports such an API at this point.
> The experiments performed serve as examples of how to apply predicted variables and to demonstrate that they are a viable solution to enable software developers to add ML models into their regular development workflow at a low engineering cost.
> Arguably, the current state of machine learning does not yet make "ML as easy as if statements" which is why we removed that claim from our paper.

---

### Official Review · AnonReviewer2 · 2018-11-03
**Potentially interesting idea, not well explained and justified**

**Rating:** 5
**Confidence:** 3

**Review:**

This paper proposes using predicted variables(PVars) - variables that learn
their values through reinforcement learning (using observed values and
rewards provided explicitly by the programmer). PVars are meant to replace
variables that are computed using heuristics.

Pros:
* Interesting/intriguing idea
* Applicability discussed through 3 different examples

Cons:
* Gaps in explanation
* Exaggerated claims
* Problems inherent to the proposed technique are not properly addressed, brushed off as if unimportant

The idea of PVars is potentially interesting and worth exploring; that
being said, the paper in its current form is not ready for
publication.

Some criticism/suggestions for improvement:

While the idea may be appealing and worth studying, the paper does not address several problems inherent to the technique, such as:

- overheads (computational cost for inference, not only in
  prediction/inference time but also all resources necessary to run
  the RL algorithm; what is the memory footprint of running the RL?)

- reproducibility

- programming overhead: I personally do not buy that this technique -
  at least as presented in this paper - is as easy as "if statements"
  (as stated in the paper) or will help ML become mainstream in
  programming. I think the programmer needs to understand the
  underpinnings of the PVars to be able to meaningfully provide
  observations and rewards, in addition to the domain specific
  knowledge. In fact, as the paper describes, there is a strong
  interplay between the problem setting/domain and how the rewards should be
  designed.

- applicability: when and where such a technique makes sense

The interface for PVars is not entirely clear, in particular the
meaning of "observations" and "rewards" do not come natural to
programmers unless they are exposed to a RL setting. Section 2 could
provide more details such that it would read as a tutorial on
PVars. If regular programmers read that section, not sure they
understand right away how to use PVars. The intent behind PVars
becomes clearer throughout the examples that follow.

It was not always clear when PVars use the "initialization function"
as a backup solution. In fact, not sure "initialization" is the right
term, it behaves almost like an "alternative" prediction/safety net.

The examples would benefit from showing the initialization of the PVars.

The paper would improve if the claims would be toned down, the
limitations properly addressed and discussed and the implications of
the technique honestly described. I also think discussing the
applicability of the technique beyond the 3 examples presented needs
to be conveyed, specially given the "performance" of the technique
(several episodes are needed to achieve good performance).

While not equivalent, I think papers from approximate computing (and
perhaps even probabilistic programming) could be cited in the related
work. In fact, for an example of how "non-mainstream" ideas can be
proposed for programming languages (and explained in a scientific
publication), see the work of Adrian Sampson on approximate computing
https://www.cs.cornell.edu/~asampson/research.html
In particular, the EnerJ paper (PLDI 2011) and Probabilistic Assertions (PLDI 2014).

Update: I maintain my scores after the rebuttal discussion.

---

> ### Author Response · Authors · 2018-11-19
> **Added reproducibility data and incorporated feedback in paper**
>
> We thank the reviewer for relevant and insightful comments. We provide responses and, when applicable, pointers to the changes we’ve done in the paper aiming to address some of the problems related to the technique we introduced.
>
> - computation overhead
> We did not provide an analysis of the computational overhead of our method because we see the three algorithmic problems as tasks to demonstrate that the interface that we provide is expressive and powerful enough to bring ML into normal software development. In many other applications, where predicted variables can be applied, speed is not a relevant metric, e.g user modelling, optimizing UI components, predicting user preference,  systems optimization, or content recommendations. We acknowledge that our current implementation is probably slower than the original variant - but as we describe above, we don't consider actual runtime to be the relevant metric here.
> Further - we strongly believe that specialized hardware such as GPUs or TPUs are continuously improving the runtime of ML models which will eventually make our proposed implementation practical even for speed sensitive applications (compare also Kraska et al, 2017).
>
> - reproducibility
> We acknowledge that the paper does not provide sufficient data related to reproducibility and we present additional reproducibility experiments in the appendix. Similar to other RL work, there are some problems with reproducibility. However, for binary search we obtain positive results (negative cumulative regret) with a reproducibility of 85% (Quicksort: 94%).
>
> - applicability
> We assume throughout our work that the developer -- algorithm and problem expert -- has domain-specific knowledge that is relevant for the problem being solved. Therefore our interface enables the developer to make use of their expert knowledge without requiring deep machine learning expertise. The developer decides what are the most important contextual signals and what metric to optimize for - The API naturally translates these into observations and rewards for the RL methods applied.
>
> - initial function
> We thank the reviewer for pointing out the lack of more detailed explanations. The initial function does not serve only for initialization but it plays two other important roles
> (1) it generates safe experience trajectories from which the off-policy RL algorithm learns and
> (2) can be reused as a safety net, should the model performance degrade.
> We have updated our draft to more clearly express this.
>
> - performance/episodes
> We are not 100% sure what the reviewer means with the comment about "performance" - we try to respond to this comment as good as we can.
> As we describe in the paper, we measure cumulative regret as our main performance metric. A negative cumulative regret indicates that the user benefits from using a predicted variable compared to the baseline. While initially, the predicted variable might perform a bit worse than the baseline, the goal is to outperform the baseline as quickly as possible. Note also, that the use of the initial function in our setup enables us to ensure a certain safety net in the beginning which helps the method to never perform terribly badly.
>
> - citations, related work
> Thank you for the reference, we have updated our draft to point out work related specifically to approximate computing, as well as for probabilistic programming.

---

### Meta-Review · Area_Chair1 · 2018-12-18
**innovative idea, contributions insufficient**

**Confidence:** 4
**Recommendation:** Reject

**Metareview:**

The paper proposes a framework at the intersection of programming and machine learning, where some variables in a program are replaced by PVars - variables whose values are learned using machine learning from data. The paper presents an API that is designed to support this scenario, as well as three case studies: binary search, quick sort, and caching - all implemented with PVars.

The reviewers and the AC agree that the paper presents and potentially valuable new idea, and shows concrete applications in the presented case studies. They provide example code in the paper, and present a detailed analysis of the obtained results.

The reviewers and AC also not several potential weaknesses - the AC will focus on a subset for the present discussion. The paper is unusual in that it presents a programming API rather than e.g., a thorough empirical comparison, a novel approach, or new theoretical insights. Paper at the intersection of systems and machine learning can make valuable contributions to the ICLR community, but need to provide a clear contributions which are supported in the paper by empirical or theoretical results. The research contributions of the present paper are vague, even after the revision phase. The main contribution claimed is the introduction of the API, and that such an API / system is feasible. This is an extremely weak claim. A stronger claim would be if e.g., the present approach would advance the state of the art beyond an existing such framework, e.g., probabilistic programming, either conceptually or empirically. I want to particularly highlight probabilistic programming here, as it is mentioned by the authors - this is a well developed research area, with existing approaches and widely used tools. The authors dismiss this approach in their related work section, saying that probabilistic programming is "specialized on working with distributions". Many would see the latter as a benefit, so the authors should clearly motivate how their approach improves over these existing methods, and how it would enable novel uses or otherwise provide benefits. At the current stage, the paper is not ready for publication.